# A Methodological Framework for Developing a Smart-Tourism Destination in the Southeastern Adriatic–Ionian Area

**Valentina Ndou** [1,*] , **Eglantina Hysa** [2] **and Ylenia Maruccia** [1]

1 Department of Engineering for Innovation, University of Salento, 73100 Lecce, Italy
2 Department of Economics, Epoka University, 1032 Tirana, Albania
* Correspondence: valentina.ndou@unisalento.it

**Abstract:** This paper presents a methodological framework aiming to support the creation of a smart-tourism destination. Specifically, the study is realised in the frame of NEST, a European Union Interreg project aiming to create a smart-tourism destination in the Adriatic–Ionian area. Therefore, the study focuses on the southeastern Adriatic–Ionian area, as a portion of the European macro-region including the Italian regions of Molise and Apulia and the Balkan countries of Albania and Montenegro. This area presents a clear vocation for tourism, with a distinctive cultural heritage, attractive natural landscapes, and old culinary traditions offering a variety of eno-gastronomic and folk craft products. In the frame of the EU neighbour policies and in coherence with the objectives of the EU smart specialization strategy, several initiatives have been promoted for innovating the tourism offering in this area. Despite this, the full achievement of the creation of an Adriatic–Ionian smart destination calls for the adoption of a multidimensional strategy able to leverage knowledge-intensive dynamics of collaboration. This paper, therefore, aims to highlight the opportunities of adoptions and implications of this methodological framework for the cross-border marketing and management of the Adriatic–Ionian smart destination.

**Keywords:** smart-tourism destination; smart specialisation strategy; cross-border tourism; methodological framework; Adriatic–Ionian tourism; sustainable development; value co-creation

## 1. Introduction

In recent years, the European Union (EU) has reserved particular attention for the policies of cooperation with the launching of several cross-border projects aimed to promote the integration and enlargement of states' membership [1]. These policies have aimed to promote inclusion and the reinforcement of socioeconomic conditions into the EU, and in the meantime, to overcome strong regionalization tendencies and the political collapses registered in several countries of Central and Eastern Europe [2,3]. Accordingly, the EU dedicated several financial instruments for the development of the projects of cohesion and prosperity in the neighbour areas, by reserving particular attention for the specialization and vocational attitudes of regions [4].

In this scenario, tourism has become one of the most appropriate sectors for the development of cross-border development initiatives. It has been widely argued that cross-border tourism development contributes to enhanced destination competitiveness by advancing local knowledge that encompasses learning and the exchange of knowledge [5,6].

The development of cross-border tourism strategies needs to be based on identification, evaluation, and definition of the similarities at the regional, sectorial, and managerial level, which impact the type and nature of knowledge sharing and innovation diffusion among participating actors [5]. From this perspective, the EU strategy of smart specialization provides relevant guidelines. Highlighted by the European Commission as a central pillar of the Europe 2020 Strategy, smart specialization is a strategy based on the identification of the regional vocation of each area, to follow a development path by leveraging specific

key enabling technologies and promote structural change [7,8]. Study of Weidenfeld [9] has identified three possible diversification strategies to be followed: (1) diversification across related tourism subsectors (intra-industry); (2) diversification across tourism and other sectors (interindustry); (3) tourism as a catalyst across other nontourism sectors. This process assumes more importance in a cross-border paradigm, when dealing with countries having multidimensional aspects to leverage in terms of geographical location, natural landscape, culture, traditions, managerial processes, and infrastructures [10,11].

Moreover, when dealing with the development of tourism strategies, one cannot ignore the adoption, diffusion, and absorption of technologies and ICT in the tourism sector, processes for which both concepts of smart tourism and smart-tourism destinations have been introduced in the literature in recent decades, where the latter are special cases of smart cities and apply smart-city principles to urban or rural areas, and not only consider residents but also tourists in their efforts to support mobility, resource availability and allocation, sustainability, and quality of life/visits [12,13].

Therefore, in order to define a cross-border strategy for the development of a smart cross-border tourism destination, it is fundamental to embrace methodological approaches for the preliminary analysis of the different countries to find out the similarities, differences, and peculiarities of each region; to cross-correlate these factors in tourist-integrated paths; and to promote the development of ICT infrastructures and knowledge-intensive services [14].

To our knowledge, this is the first study of proposing a conceptual methodological framework that integrates different levels of analysis at a macro-, micro-, and customer-level on different features that characterize a smart-tourism destination, which evaluates its promptness to adopt a smart configuration. Furthermore, this framework can be considered fundamental to constructing a strategic roadmap for the development of a smart destination. Lastly, this study is considered important since it considers a particular zone which is a recognized tourism region, that of the Adriatic–Ionian area.

Framed on the above premises, this paper presents a conceptual methodological framework that integrates different levels of analysis on several dimensions characterizing a tourism destination, in order to assess and evaluate the state of the art of a tourism destination in terms of readiness to take useful actions for a smart evolution.

The cross-border area taken in consideration is the southeastern Adriatic–Ionian area that is one of the areas in which EU funding (such as the IPA CBC programme) has been issued with the aim to develop innovative strategies for enhancing competitiveness.

Thus, this paper will demonstrate, first of all, how our methodological framework plays a crucial role in order to assess if there is awareness of the current trends emerging in tourism, especially within the smart configuration of tourist destinations, and how much such smart configuration is integrated within the regional/national system. Secondly, this paper aims to demonstrate how the methodological framework allows pointing out the strategies, policies, and features of each country with the final aim of building a cross-border tourism brand by leveraging the similarities, differences, and peculiarities of this area.

## 2. Background

Recently, smart tourism has become a buzzword among both academics and practitioners [14] and it has been largely adopted as a concept to highlight the increasing dependence of tourism destinations from the adoption and application of digital technologies to grasp huge amounts of info and data that could be transformed into value propositions [12,15–17].

Smart tourism is defined as the type of tourism that requires the integration of many factors and components such as physical infrastructure, social connections, state/organization resources, human mind, and environmental awareness [18,19]. For instance, also taking into account the crisis of COVID-19 and the effects of the tourism sector [20] , the study by [21] emphasizes the urgent need for institutions' and governmental bodies' intervention to respond to climate change problems comprising smart strategies, particularly in urban areas with regard to tourism.

On the other hand, a tourism destination is comprised of different inter-related stakeholders, forming, thus, a kind of cluster that interacts in a social network with the aim to realise a tourism experience that is aligned with visitor needs [22]. The advent of digital technologies has significantly impacted the tourism sector. This could be evinced from the technological trend that characterizes the sector, where a significant growth of new distribution channels, virtual communities for tourism, and social media platforms have been registered that support tourists to make smart decisions [23,24].

On the other hand, the adoption of mobile technology, the internet of things, and wearable devices has contributed to enhancing tourists' empowerment, thus transforming the tourism experience [25,26]. Moreover, the co-creation experience has emerged as a new phenomenon for tourists to create new value for their experiences [27–29].

The adoption of such new advanced technologies in tourism has created the foundation for the birth of the smart-tourism destination (STD) where technology becomes an enabler and a vital driver for the competitiveness of a destination [30]. Advanced services, a high degree of innovation, and the presence of open, integrated, and shared processes for enhancing the quality of life for both residents and tourists are the essential features of an STD [31]. According to [32], the STD concept arises from the advancement of smart cities.

One of the most relevant definitions of an STD available in the literature is the one provided by [33], that defines it as:

*"An innovative tourist destination, built on an infrastructure of state-of-the-art technology guaranteeing the sustainable development of tourist areas, accessible to everyone, which facilitates the visitor's interaction with and integration into his or her surroundings, increases the quality of the experience at the destination, and improves residents' quality of life".*

Ref. [34] conceptualized the STD according to five main layers that span from the physical layer to technology up to the experience layer (Figure 1).

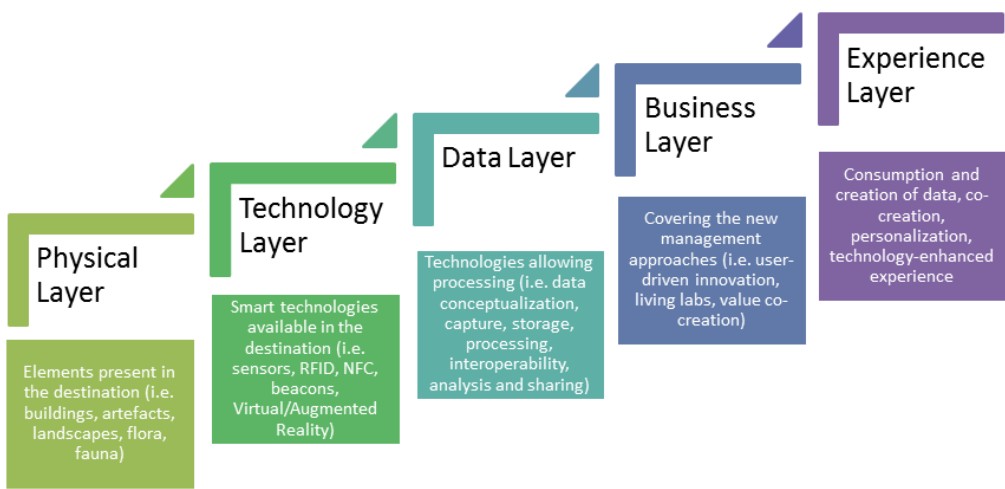

**Figure 1.** Conceptual Layers of Smart Tourism (adapted from [34]).

According to [34], these five layers need to be evaluated at the destination level to understand to what extent the STD is developed in the area. Therefore, this framework could be a basis for the analysis of the smartness of a destination at different layers and levels.

In addition to this, in order to provide a generic framework for smart destinations, [35] enlarged the STD concept to also incorporate in it competitiveness, sustainability, and inclusiveness. It is worth noting that ICT, people (intended as human capital, social capital, and knowledge management), and leadership (meant as a participatory government, policies and regulations, and change management), if intertwined and interconnected within the smart-ecosystem process, can become contributors to the smartness concept and should, therefore, be enhanced and developed to co-create innovation [30,36,37].

For all these reasons, in order to work toward the development of a smart and sustainable tourism destination, it is necessary to assess the "state of the art" of a tourism destination under different perspectives and evaluate its promptness to adopt a smart configuration.

The different research realized up to now has analysed and evaluated many aspects of smart tourism and STDs, with the aim of grasping the importance of smart-tourism technologies' characteristics, or the information quality, interactivity, and accessibility, and how all these influence travellers' decision-making processes and thus lead to their travel-decision-support satisfaction; investigating tourists' preferences of smart tourism quantitatively in a tourist-attraction context and give useful directions for the diagnosis of strengths and weaknesses of smart-tourist-attraction construction [38]; identifying and clarifying the employability-skill deficits in rural hospitality and tourism destinations [39]; and understanding how smart cities may foster collaboration ecosystems that may improve both the standards of living and the competitiveness of urban spaces [40,41]. Nevertheless, no studies have shown how to evaluate the degree of smartness of a destination.

Due to the lack of a comprehensive framework that puts all these different elements, variables, and features together, in this paper, we aim to close this gap by setting up a conceptual methodological framework that integrates different levels of analysis on the different dimensions characterizing a tourism destination. The methodological framework could also be relevant for assessing and evaluating the state of the art of a tourism destination in terms of smartness.

*Relevance for Adriatic–Ionian Area*

The southeastern Adriatic–Ionian area is a culturally diverse macro-region that has unique cultural heritage, attractive natural landscapes, and old culinary traditions offering a variety of eno-gastronomic and folk craft products. According to the Interreg Mediterranean Report [42], tourism is one of the most important drivers of this area's economy, in terms of absolute value, gross value added (GVA), and employment. Nevertheless, this area is characterized by inefficient cross-border territorial synergies, seasonal tourism demand, a lack of brand reputation, an absence of sustainable-identity promotion strategies, and difficulties in accessibility. This is due to the vocation of the area and also to the cross-sectorial nature of the tourism industry. All this moves toward the adoption of a cross-border and multistakeholder approach able to leverage the multitude of resources and factors belonging to the macro-area.

Specifically, at a regional level, many countries are implementing important tourism-development strategies in this sector, with the aim of accelerating socioeconomic development through investments aimed at generating technological innovation [43] and the knowledge society, adaptability to economic and social changes, protection and improvement in the quality of the environment, and administrative efficiency. On the other side, the Western Balkan economies, with the aim of accession to the EU, have introduced smart specialization as a way to facilitate economic growth and to align policy priorities across sectors with support from the European Commission's Joint Research Centre [44]. As the current challenges in tourism development include the need to upgrade tourism infrastructure and to enhance public–private cooperation [45], introducing smart specialization to the Western Balkans provides an opportunity to tackle them [46]. Both Albania and Montenegro offer an example of how to do so from a cross-sectorial perspective based on related variety. From a first qualitative analysis of these territories, it emerges that all these countries are characterized by the "triple S": sun, sand, and sea. Their coastlines are rich in diversified coasts and alternating beaches, while inland, they exhibit different peculiarities due to their multitude of natural and cultural resources. In particular, Pedrini's [47] definition of a region with a tourism vocation with its combination of favourable environmental factors, cultural heritage, agro-food, and accessibility is well-suited to the four countries [14,46,48–52].

By exploiting and bringing together both appealing similarities and differences, combined with the variety of their peculiarities, the final aim is to build a common brand of the southeastern Adriatic–Ionian area as an inter-regional STD and develop common tourist experiences, products, and services. To this aim, a methodological framework has been proposed with the intention of evaluating and grasping similarities and differences in order to determine the best innovative strategy for cross-border tourism development in the area.

## 3. Methodological Framework

Our framework (Figure 2) rests its foundations on the conceptual layers that define an STD [34] and embraces the idea that the interconnection and close collaboration between leadership, people, and ICT can lead to the development and growth of a smart territory [30,36,37].

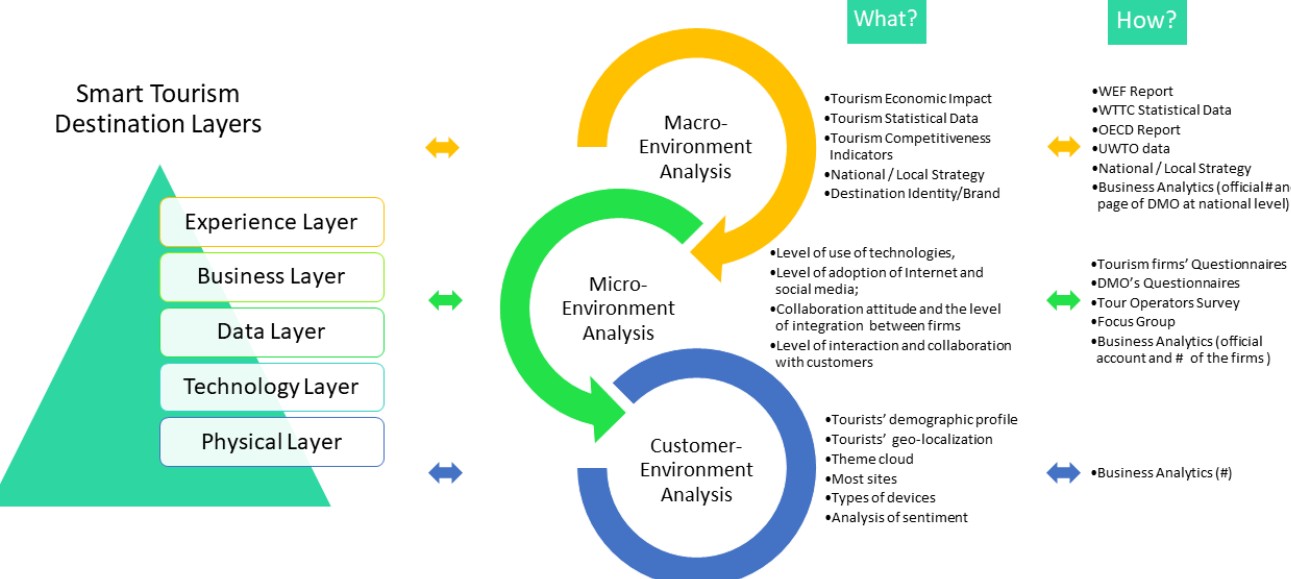

**Figure 2.** Methodological framework.

The framework aims to assess the readiness of a destination to be smart and to improve its services and offers three main pillars: the macroenvironment, the microenvironment, and the customer experience analysis.

More specifically, the macro-perspective analysis consists of the investigation of all the characteristics of a destination, the economic impacts of tourism, the level of competitiveness of the destination, and its brand or identity, with the final aim of discovering the major trends and opportunities for the development of strategies for the marketing and management of a tourism destination. Moreover, the micro-perspectives focus their attention on the tourism firms' competitiveness, their technological promptness, and the diffusion/adoption of ICT tools, while the customer perspective focuses on analysing customers' profiles, understanding and anticipating their needs, personalizing tourism offers, co-creating the tourism experience, and evaluating customer satisfactions. This assessment is realised through the use of big data analytics to process data generated on online social networks [53,54].

### 3.1. Macro-Perspective Analysis

The macro-dimension analysis part of the framework aims to assess the tourism destination in its main characteristics related to the first four layers identified by [34], i.e., physical layers and technological infrastructure, the data layer, and business layer, by taking into consideration the impact of the tourism sector within the economy of the country. From this perspective, first of all, the macro-analysis should consider specific

key indicators, such as tourism's direct/indirect contribution to GDP and to employment, investment, and visitors' exports [55]. Besides the economic impact of the tourism sector within the country, important statistical information should be integrated when performing this kind of analysis: the touristic flows (in terms of domestic, inbound, or outbound tourism), the tourism receipts, and expenditure and other useful information about tourism enterprises are crucial, in fact, for describing trends and opportunities for the development of an STD.

Moreover, the definition of the state of the art of tourism destination needs to study the destination competitiveness, defined as the ability of destinations to deliver better tourist experiences and to create a better living environment for local residents and other destinations [56]. Buhalis, D. [57] and Hassan, S. S. [58] highlight the relationship between competitiveness and economic prosperity and the delivery of an experience that is more satisfying compared to other similar destinations. Understanding country competitiveness in tourism is both a major consideration for policy makers and a major challenge for professionals, as it provides evidence in decision-making processes [59]. From this perspective, in recent years, a large variety of indicators, such as the T&T Competitiveness Index of the World Travel & Tourism Council (WTTC), considered the most complete and modern set of indicators globally available, have been developed by different organizations to address particular aspects of competitiveness. Thus, the analysis presented here can be performed through the use of the following family of indicators (more details are provided in Appendix A).

Tourism's direct, indirect, and total contribution to GDP and employment, capital investments, and visitor exports, through the exploitation of the WTTC annual reports for assessing tourism's economic impact;

Tourism flows (domestic, inbound, and outbound), receipts, and expenditures, together with tourism enterprises, through the analysis of OECD annual reports for measuring the tourism performance and the capacity of a destination to be attractive on both the national and international level;

ICT readiness, prioritization of travel and Tourism, international openness, price competitiveness, environmental sustainability, air transport infrastructure, and other indicators, through the exploitation of the T&T Competitiveness Index.

Apart from the global competitiveness indicators, through the use of business analytics, the study of a destination's brand or identity should be performed on social media platforms, as well as the identification of political tourism strategies at both the national and regional level, as they play an important role in the development of a tourism destination and its competitiveness and offer useful information related to the destination from a macro-level perspective. In particular, the focus is to see how and to what extent the institutional capacity for coordination, collaboration, and cooperation can be efficiently used as a governance practice (the efficiency of governance) to improve tourism-destination competitiveness, helping to transform tourists' needs into solutions and opportunities for smart, inclusive, and sustainable growth [60].

### 3.2. Micro-Perspective Analysis

The micro-dimension analysis of a destination aims to understand the dynamic interconnection among different stakeholders with the purpose of favouring service co-creation, service exchange, and value co-creation, through the use of social media and internet tools that enable them to network. The convergence of ICTs with the tourism experience and its enhancement through personalization, context awareness, and real-time monitoring make smart technologies crucial both in the development of an STD and of the final destination image [23,61]. This analysis involves different layers such as those by [34], from the physical to the business one, passing through adopted technologies and data layers, and it is strictly related to the final destination image.

In order to perform this analysis, a field research methodology could be used for grasping the detailed performance of the emergent features of an STD from a micro-

dimension perspective. Field research consists of gathering data regarding the promptness of the location to adopt a smart-destination approach through the definition of specific surveys to be administered to different typologies of tourism players, such as hospitality firms (hotels, B&Bs, agritourists, and so on), intermediaries (travel agencies, tourist guides, and so on), destination management organizations (DMO), and so on. An example of a survey is reported in Appendix B. Each question has the objective of defining the level of smartness of the tourism in a specific territory in terms of its readiness and promptness for the adoption of digital technologies, the creation of digital local experiences, the capacity to network and collaborate, and, mostly, to evaluate the following indicators:

- The level of use of technologies, internet, and social media;
- The collaboration attitude and the level of integration with the other actors of the tourism system for creating new innovative opportunities;
- The level of interaction and collaboration with customers for experience and service co-creation.

*3.3. Customer-Perspective Analysis*

One of the most fundamental layers in the touristic field is the experiential component, which focuses on the affective and emotional component of the consumption process [34]. The concept of experience is strictly connected with the entertainment aspect: contemporary tourists want to live a unique experience and are not interested anymore in purchasing a standardized product/service [62]. Therefore, in order to meet the new needs of the demand, tourist destinations must give top priority to the achievement of tourist satisfaction.

Another aspect related to the experiential layer and the consumer satisfaction in the touristic sector regards future consumer intentions, loyalty, and word-of-mouth communications [63]. The use of data gathered on social media, blogs, forums, and so on facilitates touristic firms to know the needs of tourists and to plan some processes, such as marketing and sharing just-in-time information about attractions, catering facilities, transportation alternatives, and so on [64]. Using the insights gained from big social data, defined as "that subset of Big Data generated from people's actions and interactions within social media services and platforms, properly collected and analysed to provide crucial insights into human behaviour, people's preferences and relationships, social interactions and transformations, and real-life outcomes prediction" [65], it is important in the tourism field to uncover new opportunities for the business to make decisions on the basis of numbers and analysis rather than anecdotes, guesswork, intuition, or past experience. From the point of view of the tourists and in terms of mobile technology, having reliable real-time information always available is crucial in terms of enabling them to find their way [12].

For all these reasons, big data analytics strategies are indispensable from a customer-perspective analysis, as they enable ascertaining consumer trends, travel patterns, threats, and opportunities. In particular, some of the enhancements that big data analytics could bring for customer experience and business efficiency improvement are: personalizing the customer experience; helping travel companies to create a better pricing strategy and customer analytics; and improving services, marketing, and sales optimization.

The micro-perspective and customer analysis are the most innovative pillars of the analysis as they allow going deeper into the details thanks to the use of both field analysis and business analytics strategies.

The results of the analysis obtained through all the three perspectives will provide a strategic basis for defining the best innovative path for the development of an STD of the cross-border area.

**4. Results**

*4.1. Macroanalysis*

Data from WTTC annual reports (https://wttc.org/Research/Economic-Impact, accessed on 25 October 2022) show that tourism is one of the sectors that contributes the most

to the total economy of each country: in 2019, for Albania, the contribution of travel and tourism to GDP was 21.2% of the total economy, with a growth in tourism GDP of 8.5% in that year and a contribution of 22.2% to total employment. For Italy, the contribution of travel and tourism to GDP was 13.0% of the total economy, with a growth in tourism GDP of 2.2% in that year and a contribution of 14.9% to total employment. For Montenegro, the contribution of travel and tourism to GDP was 32.1% of the total economy, with a growth in tourism GDP of 6.1% in that year and a contribution of 32.8% to total employment. These data confirm the positive trend in tourism growth in the Balkans during the last few years after experiencing a stalemate in the past decade, while in Italy, tourism continues to make an important contribution to its economy. Analysing these countries from a competitive perspective through the use of travel and tourism competitiveness indicators (TTCIs) (https://reports.weforum.org/travel-and-tourism-competitiveness-report-2019/country-profiles/, accessed on 25 October 2022), it can be noticed that the three countries are very different among them. Italy occupies the 8th position (to 140) with an overall score of 5.1, Montenegro the 67th with an overall score of 3.9, and Albania the 86th with an overall score of 3.6 (reference year: 2019). In Figure 3, an overview of TTCIs shows the actual situation for the three countries. The three countries can be considered as countries having a good level of safety and security, but Italy has an easier way of transport (by air or ground and port) than Albania and Montenegro, and these aspects are important for the final image of a destination. Moreover, they show similar values for environmental sustainability, price competitiveness, ICT readiness, and prioritization of T&T, thus giving the idea that it is possible to intertwine a common canvas for common strategies and services from the perspective of a smart cross-border tourism destination. Nevertheless, Italy shows higher values of international openness; thus, both Albania and Montenegro should intervene more in this direction in order to fill this important existing gap.

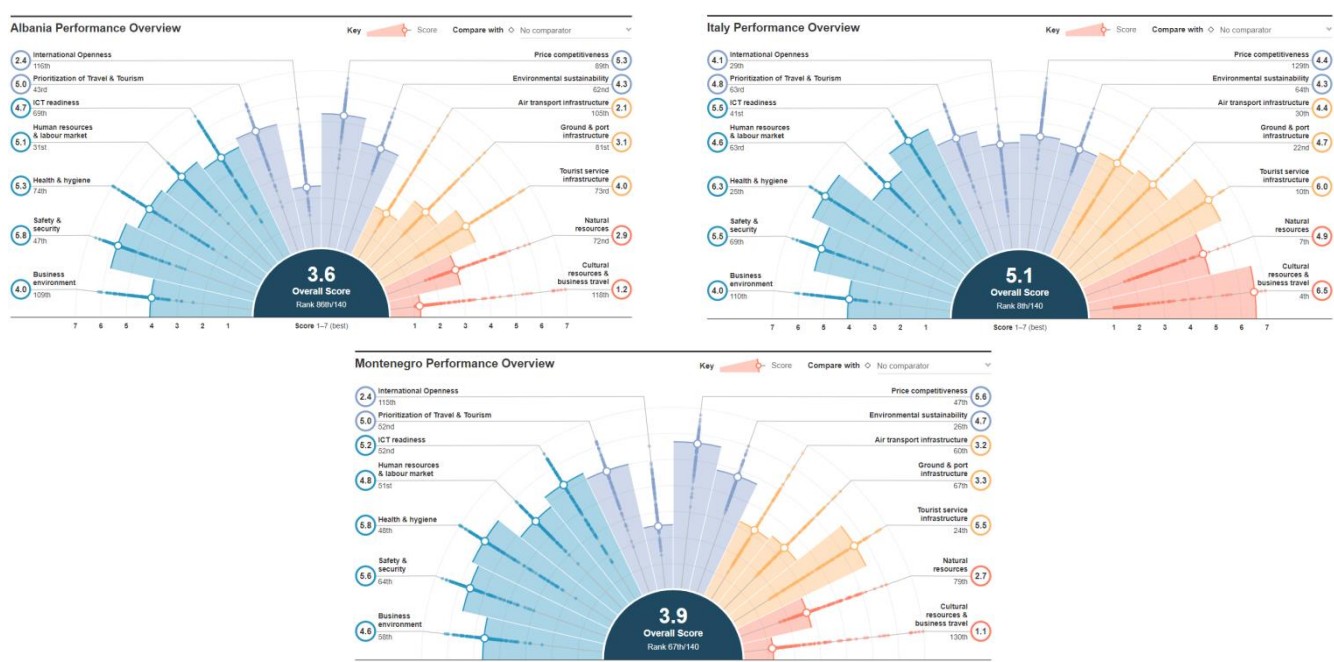

**Figure 3.** TTCI profiles for Albania, Italy, and Montenegro.

On the other hand, as described in the previous paragraphs, all the three countries have a strategic plan for the development of tourism within the national territory, and show a solid destination identity/brand, which comes from the analysis of official accounts of the destinations. In particular, they have several active social media accounts, always updated about the upcoming events and with a destination presentation, through the continuous publishing of interesting and catchy contents. These aspects assume more

and more importance for the demanding users, who are always in motion and who want to have access to information anytime and anyplace. Moreover, an online campaign on social media is active throughout the year and aims to make the brand of these countries attractive, omnipresent, and recognizable and to depict them, separately, as a unique destination whose rich offerings suit many tastes and meet expectations for all tourist profiles. In fact, through the analysis of Trip Advisor countries' showcases, a list of the top best places to visit has been compiled. Additionally, for all these regions, there are many activities proposed to the tourists that take into account most of the tourists' needs, from outdoor activities (i.e., hiking/biking, shore excursions, and tourist tours), to cultural and themed tours (historical and heritage, literary and artistic, archaeological, eno-gastronomic, nightlife, and so on), to packages suitable for families, to private and personalized tours, to one-day/private trips, and finally to multi-day tours.

*4.2. Microanalysis*

As aforementioned, a field research methodology has been adopted within the micro-dimensional investigation through the administration of tailored questionnaires to different typologies of tourism actors. Among the different issues, the questionnaires are aimed at measuring the level of use of technologies, internet, and social media by service providers, intermediaries, and so on.

A preliminary investigation of the presence of ICT infrastructure within the three countries—Albania, Italy, and Montenegro—thanks to the use of the Global Innovation Index, has highlighted that in all these countries, the ICT infrastructures are efficient. In particular, within the program countries, Italy has primacy, being at the 24th position in the ICT infrastructure pillar rank in 2019.

Before proposing innovative solutions within the countries and starting to create a smart cross-border tourism-destination strategy, it is important to understand the state of the art of the real use of technologies in all these areas. From this perspective, thanks to the questionnaires and the micro-dimension analysis, it has been possible to understand what kind of technologies are used and how much their use is widespread among the tourism stakeholders for the performance of various activities, from the basic ones to the more complex ones.

For example, the first evidence is given by the use of technologies made for the management of booking and reservation inside touristic companies. In the following panels, the difference between the four countries can be seen. In the top panels, on the left, the results for Albania are shown, while on the right can be seen the ones for Montenegro. In the bottom panels, the Italian situation, divided between Puglia and Molise, is shown (Figure 4).

It can be seen that the use of online promotional sites, such as Booking, is widespread in all the four countries, even if a small percentage of participants in Montenegro and Molise still have lower use of them than in Albania and Puglia. E-mail (for Montenegro) and telephone (for Molise) are the most common media used for booking, while two prevalent features for all the countries are on the one hand the use of a corporate web site, and on the other hand the lack of mobile apps, both as tools to promote their own activities.

In order to gain a further idea about the use of ICT, the real reasons why the internet is used by the tourist companies have been investigated. In the following panel, the difference between the four countries can be seen. On the top panels, on the left, the results for Albania are shown, while on the right can be seen those for Montenegro. In the bottom panels, the Italian situation, divided between Puglia and Molise, is shown in Figure 5.

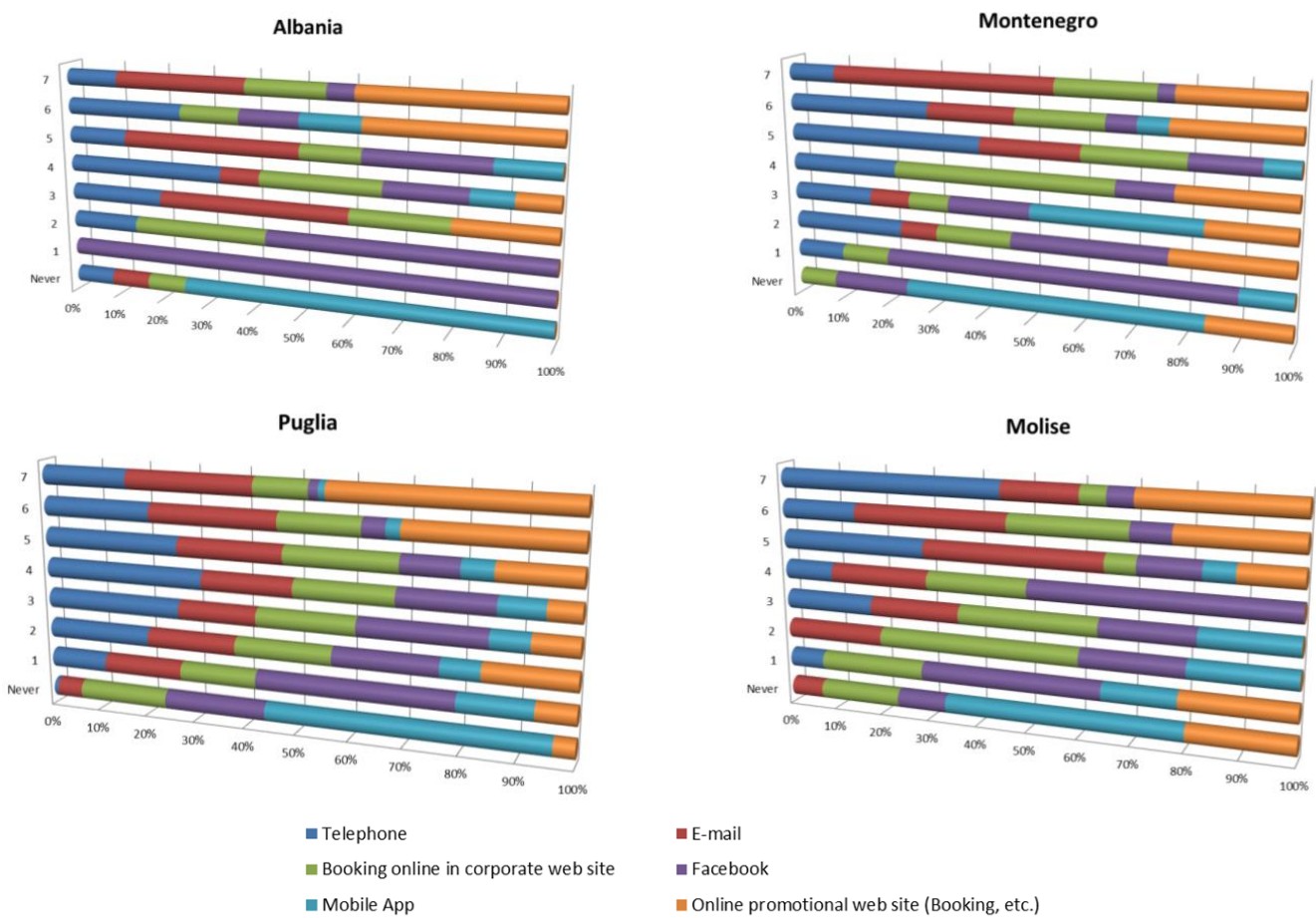

**Figure 4.** What is the extent of use of these channels for booking and reservation (1—low, 7—high)?

It can be seen that in Albania, the internet is mostly used for the management of booking and reservations, promotions, and special offerings, and for customers and fidelity initiatives. On the other hand, its use is not so widespread for the management of the relationships between industry and association categories, for communications with public administrations, and for market research. A large share of people interviewed do not use very much internet for the monitoring of customers' feedback, a practice that is considered to be important in order to improve their services and their own competitiveness at both a territorial and national level. Moreover, in Montenegro, the internet is mostly used for the management of information from customers, for the management of special offerings and promotions and, finally, for the management of fidelity initiatives, but there is a lack of its use regarding market research and buying products or services. As regards Puglia, there is a gradual growth of interest in the use of the internet for managing requests of info from customers, booking and reservations, and promotions and special offerings, together with customers' feedback. On the other hand, the use of the internet falls off for market research and for the purchase of products and services. Finally, in Molise, the use of the internet is widespread, in particular for the management of promotions and special offerings, information from customers, booking and reservations, and fidelity initiatives, while it is not so used for market research and for the purchase of products and services. It is worth noting that there is a large share of those interviewed in Molise that do not use very much internet in order to monitor customers' feedback.

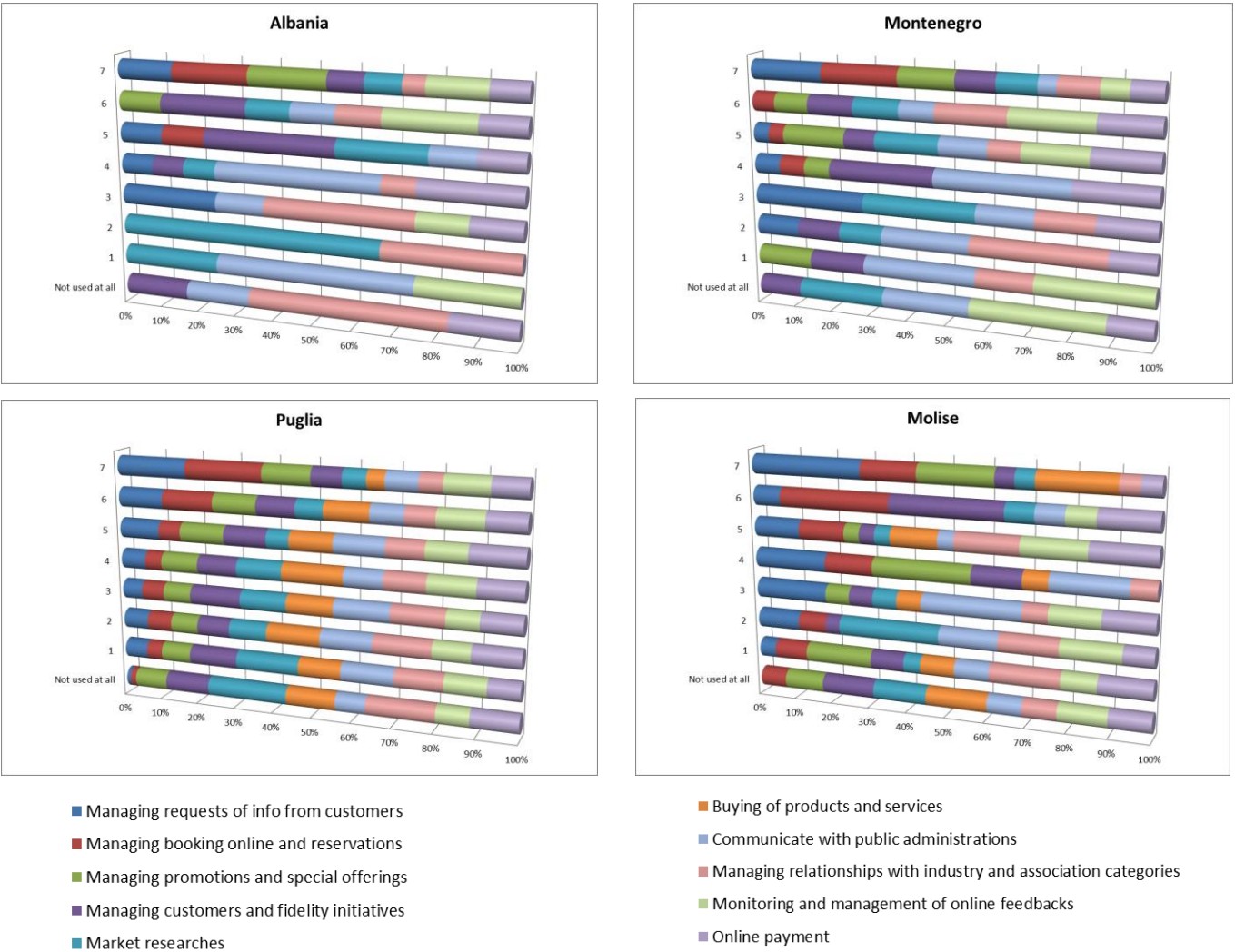

**Figure 5.** What are the reasons for the usage of the internet (1—low, 7—high)?

The diffusion of smart technologies has had an impact also on the spreading of several categories of technologies with different aims. For the purpose of our work, we have classified them into the following groups, in order to understand what their level of use is: systems for user profiling, OTA, CRM systems, social networks, blogs and forums, virtual guides, and mobile applications. In the following panel, the difference between the four countries can be seen. On the top panels, on the left, the results for Albania are shown, while on the right can be seen those for Montenegro. In the bottom panels, the Italian situation, divided between Puglia and Molise, is shown in Figure 6.

For all the countries, the most used are OTA (Booking, Expedia, and so on), followed by systems for user profiling. Then, some differences come to light, in particular for the use of CRM systems which falls down in Molise, while it is quite widespread in the other countries, and for the large adoption of social networks, blogs, and forums in the Italian regions, while in Albania and Montenegro, they seem to be in moderation.

We have seen that in all countries, despite the fact that monitoring customers' feedback is a practice that is considered to be important in order to improve services and competitiveness at both a territorial and national level, a large share of people interviewed do not use very much internet for this practice. This trend is also observable in the analysis of the tools that are mostly used in the four countries to monitor customer satisfaction. In the top panels, on the left, the results for Albania are shown, while on the right can be seen those

for Montenegro. In the bottom panels, the Italian situation, divided between Puglia and Molise, is shown in Figure 7.

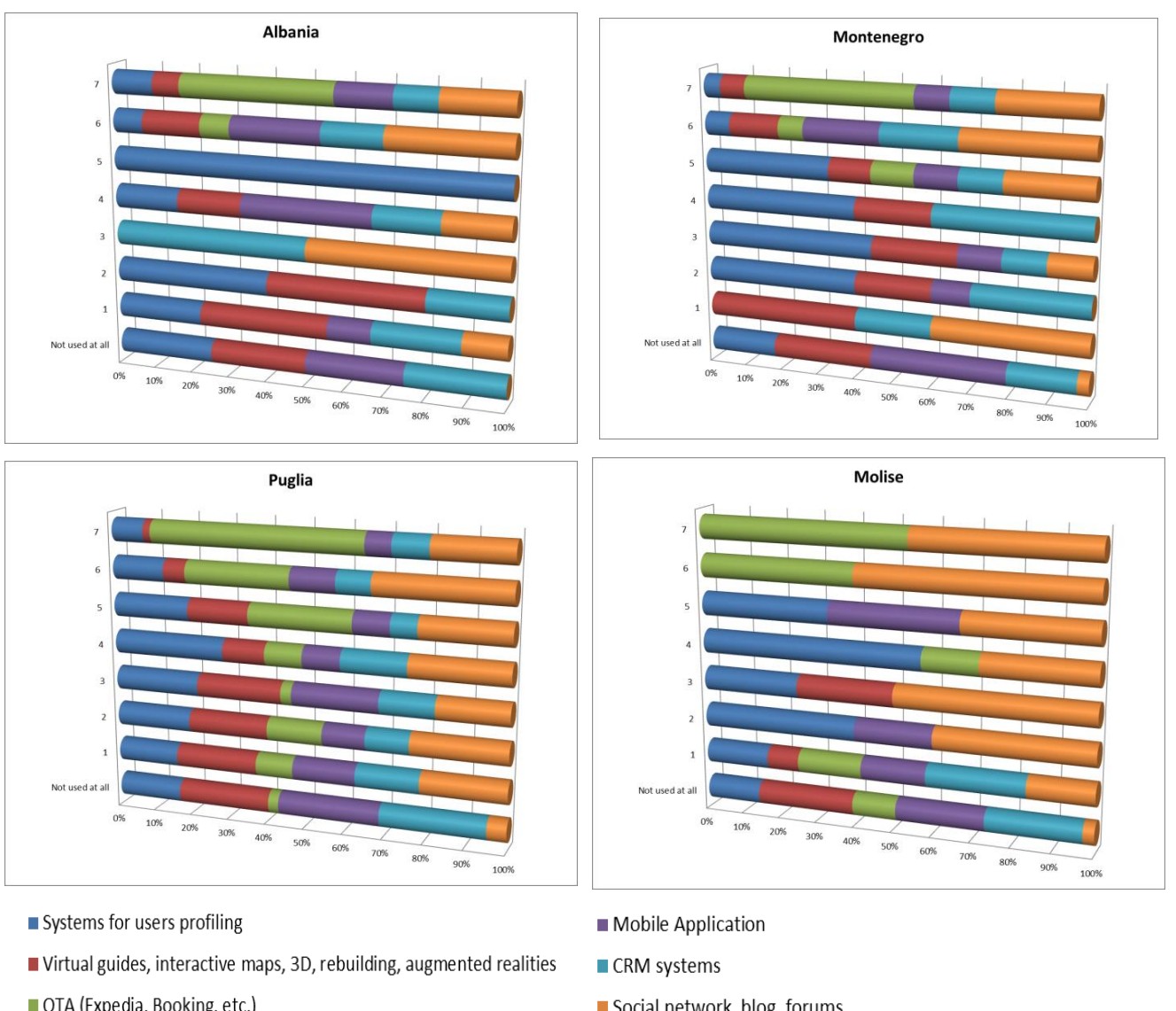

**Figure 6.** What is the level of use of the following technologies (1—low, 7—high)?

It can be seen that traditional tools, such as vis à vis, feedback/reviews, guest books, and sometimes questionnaires, are preferred to the monitoring of customer satisfaction on social networks, forums, and blogs. The last, in fact, have very high percentages among the "Not used at all" category. In a certain way, this implies that there is not much awareness of how important the data that are generated online are, and that the word of mouth on unconventional media, such as social networks and forums, can actually play a fundamental role in improving one's own competitiveness.

Nevertheless, when it was asked how data could be used, in Albania, the improvement in smart hospitality reached the highest position, together with the discovery of patterns and trends, while it had lower values for Puglia and Molise. In general, in the following panel, the difference between the four countries on how data could be used can be seen. In the top panels, on the left, the results for Albania are shown, while on the right can be seen those for Montenegro. In the bottom panels, the Italian situation, divided between Puglia and Molise, is shown in Figure 8.

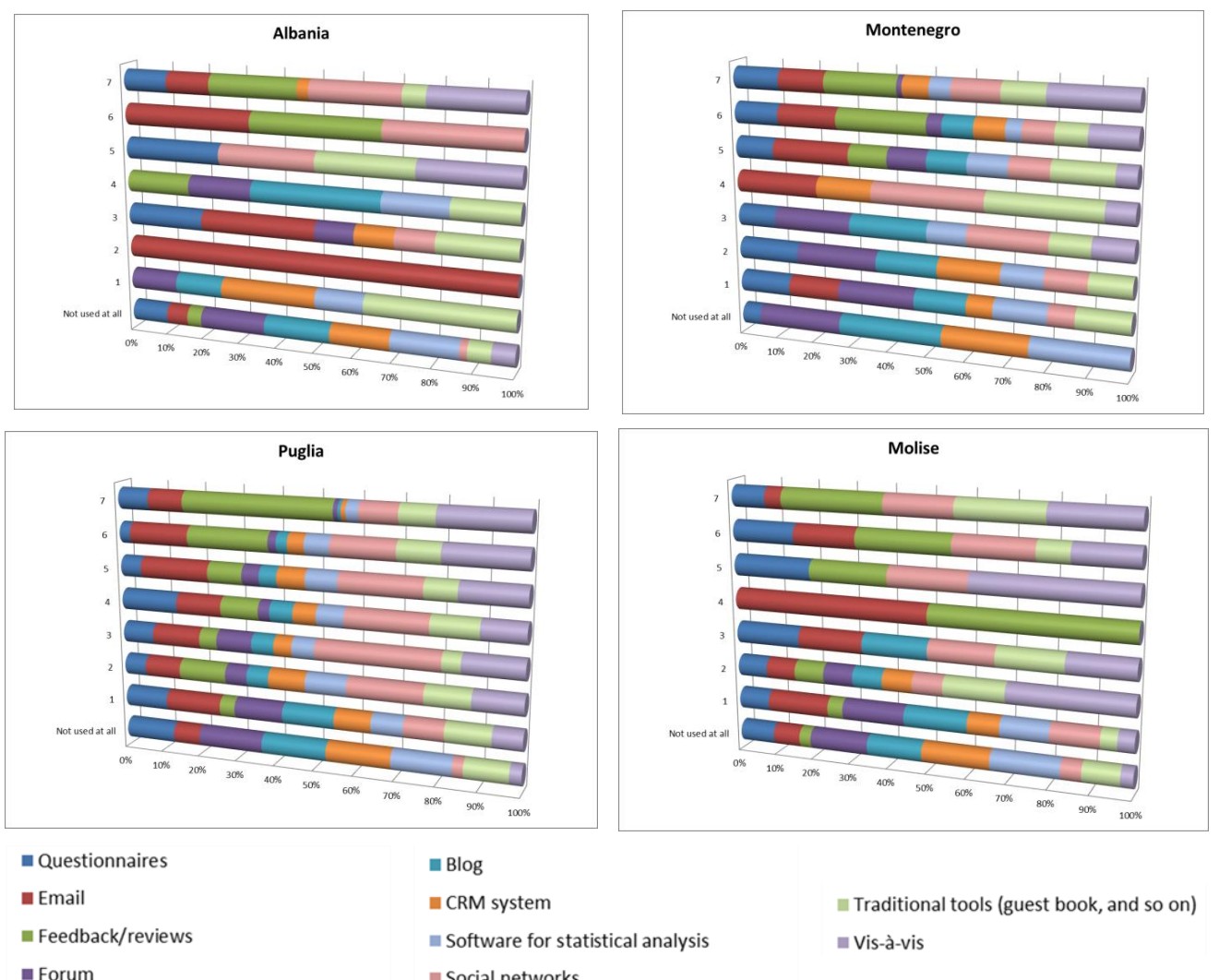

**Figure 7.** What kind of tools are used by companies to monitor customer satisfaction (1—low, 7—high).

All the options have a similar trend, with some differences in the percentage values, so it is evident that data are very useful for all the listed possibilities, from the increment of the competition power among rivals to the creation of personalized services or of marketing strategies, all aspiring to achieve an improvement in their own competitiveness.

Finally, the adoption of a mobile app is considered very useful in order to improve the competitiveness of the hospitality services by a high percentage of those interviewed among the four countries. In particular, in the following panel, the difference between the four countries can be seen. In the top panels, on the left, the results for Albania are shown, while on the right can be seen those for Montenegro. In the bottom panels, the Italian situation, divided between Puglia and Molise, is shown in Figure 9.

As can be seen, a mobile app is not considered very useful in order to improve pre-experience in all the countries, sometimes together with the local-heritage knowledge. However, it is quite evident that a mobile application is important as a facilitator for communication, for obtaining instantaneous info, and for the creation of unconventional paths/packages, sometimes together with having the use of a virtual guide, useful for the discovery of tourism destinations and for the learning of new information about them.

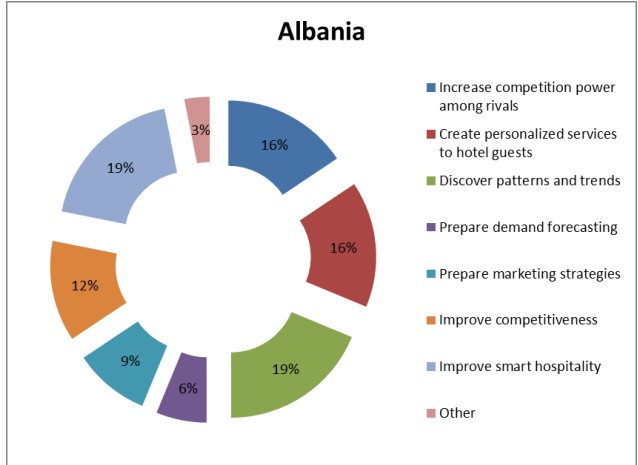
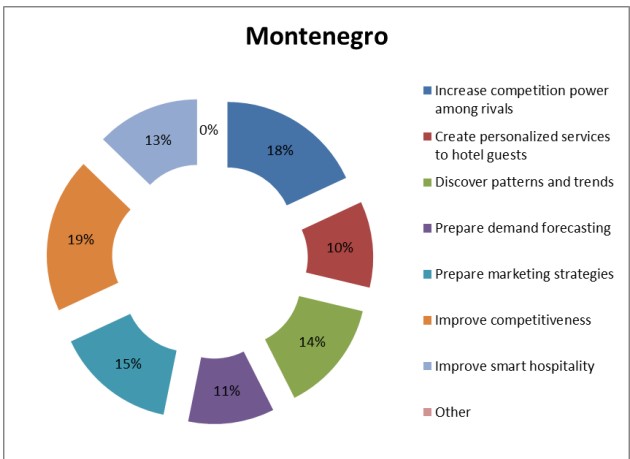
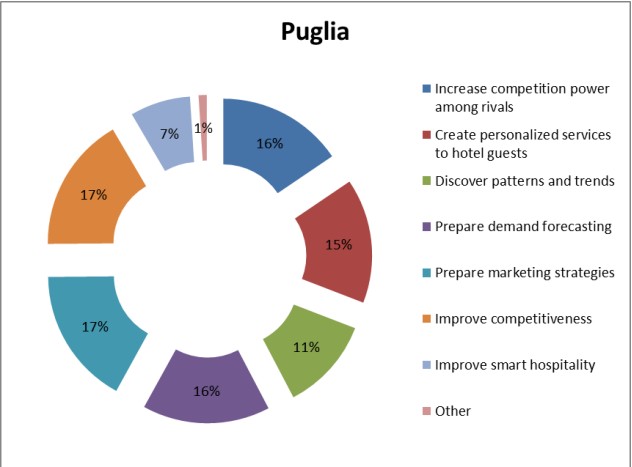
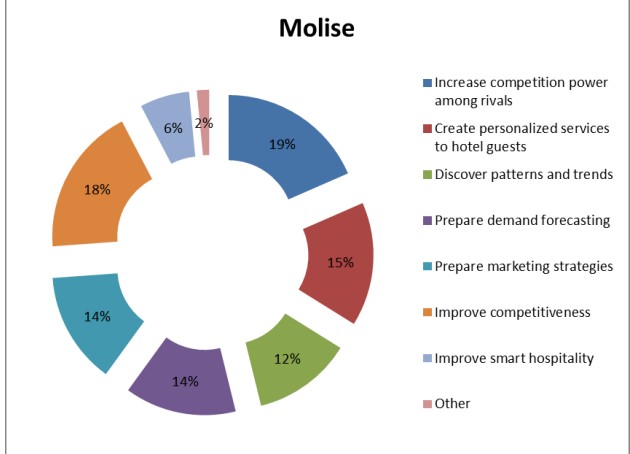

**Figure 8.** How data could be used.

## 4.3. Customer Analysis

Big data analytics has allowed understanding how and how much a country is talked about, if the experience of tourists is positive or negative, and the effectiveness of the communication through the social network used. Thanks to the use of business analytics, in fact, data from the web were gathered in order to explore the perception of the final destinations and the customer satisfaction. This kind of analysis is fundamental when the aim is that of realizing new experiential tourism products and services for visitors and of better satisfying their needs.

Four hashtags were chosen in order to gather data from the web: #albania, #molise, #weareinpuglia, and #montenegro. When gathering data, it is important to pay attention to the kind of data which we are interested in and to restrict the analysis to the field of interest. Therefore, the extraction of the data was carried out by combining the country hashtags with #tourism. In this way, a first simple selection was carried out. The tool used for the analysis was Brand24 (https://brand24.com/, accessed on 25 October 2022), which retrieves data from all of the web. In Figure 10, a first comparison between the mentions of these hashtags can be seen, showing that all of them were quite popular within social media. Nevertheless, the region of Molise should pay more attention to these channels when organizing marketing strategies for tourism promotion.

**Figure 9.** What are the areas of improvement in competitiveness of the hospitality services based on the usage of a mobile application (1—low, 7—high)?

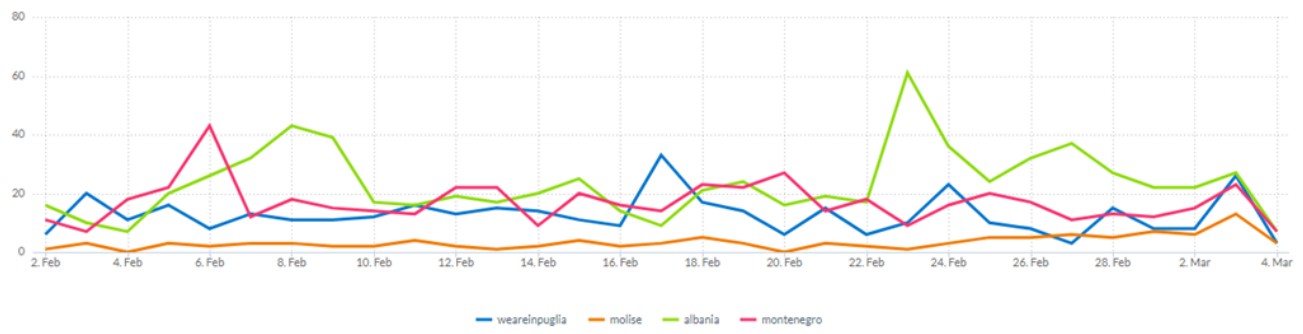

**Figure 10.** Comparison between mentions of: #Albania, #Molise, #weareinpuglia, #Montenegro, and #tourism.

A major insight is clearly visible in Figure 11, where the trends in the number of mentions, social media reach, likes, and comments by day are shown. These graphs are important because, on the one hand, it is possible to see to what extent the chosen hashtags are efficient and how much they are used. Obviously, Albania and Montenegro showed generally higher values than Puglia and Molise, as the former are at a national level.

Nevertheless, in some cases, social media interactions (i.e., social media shares, social media likes, and social media comments) had higher values for Puglia than the other countries. On the other hand, social media reach is a crucial metric as it refers to the number of users who have come across a particular piece of content on social media. Thus, even if the number of mentions was higher for Albania, it is also evident that Puglia was the region which had the best trend in social media reach and this is important for the promotion of local tourism.

Finally, in Figure 12, the overall sentiment for each dataset is shown. Positive and negative percentages show the number of contents that contains, respectively, more positive or negative keywords, and it is easy to see that people who spoke about Albania, Puglia, Molise, and Montenegro had a very positive opinion of them. Despite the fact that all the countries had very positive comments, it came to light that the region of Molise should improve something related to its marketing and services, as one mention out of five had a negative score.

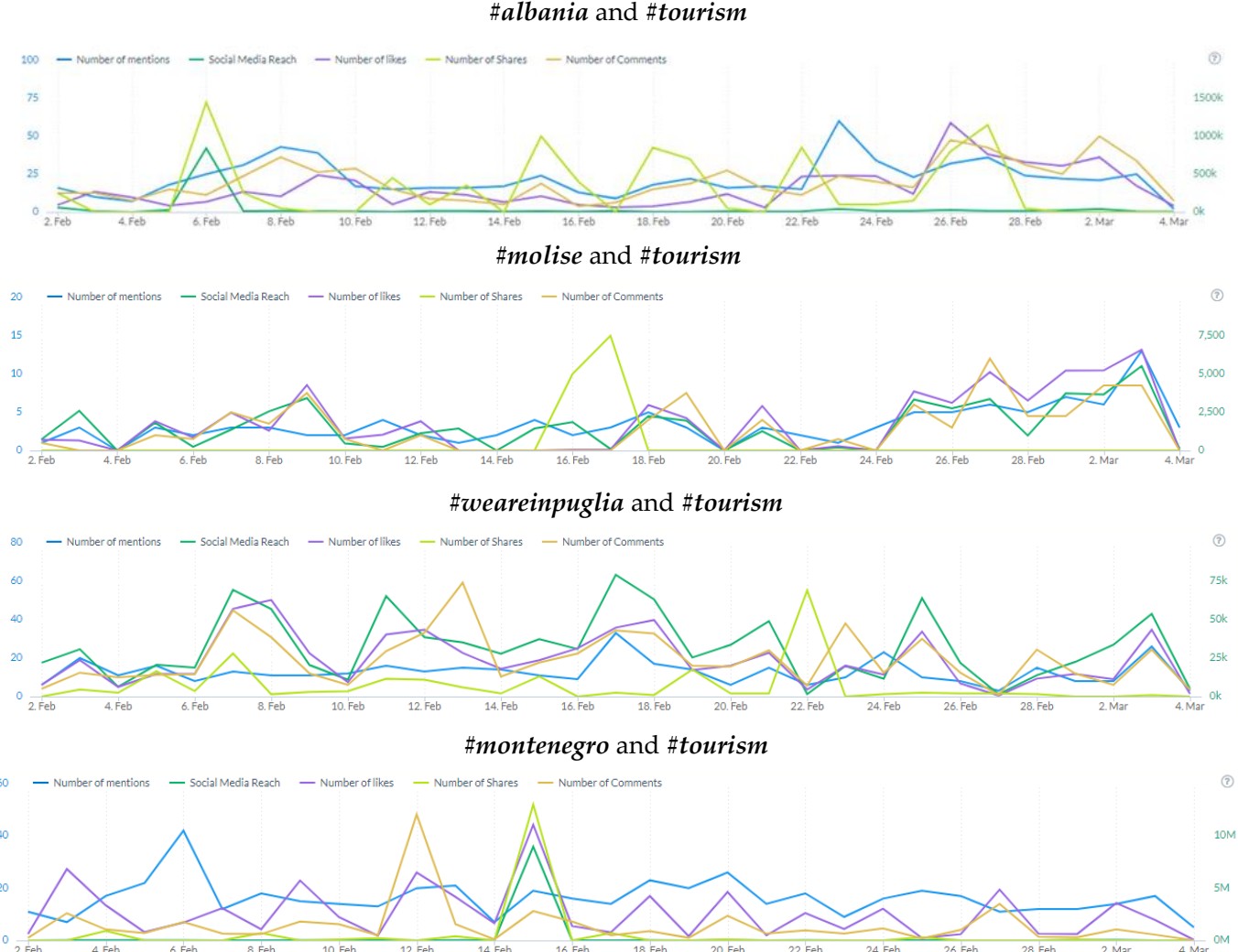

**Figure 11.** Mentions, Social Media Reach, Likes, and Comments by day for all the monitored hashtags.

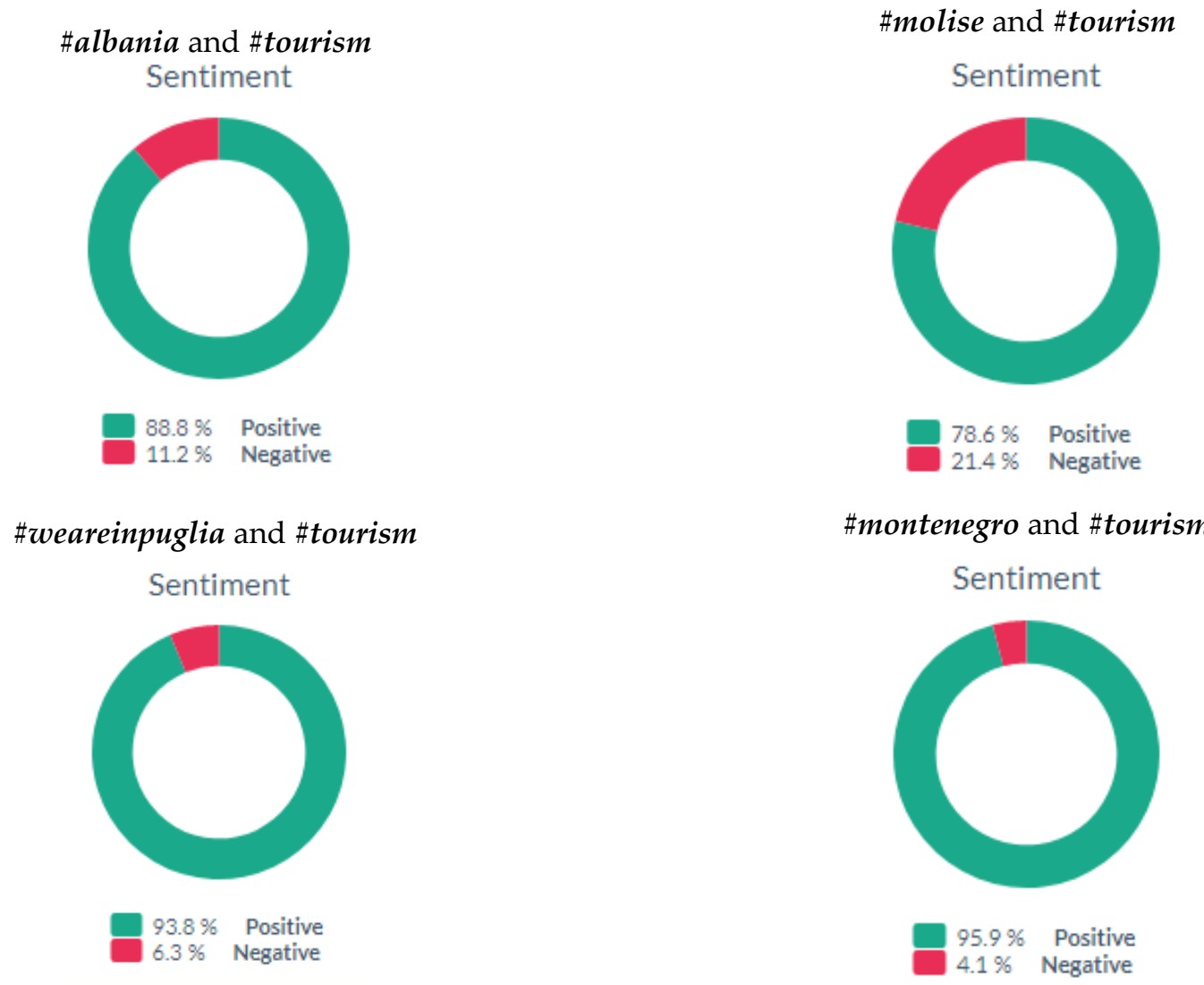

**Figure 12.** Sentiment score, divided into positive and negative, for all the mentions.

A future analysis that could be conducted is understanding the way in which the communications happen (if the posts always contain pictures, videos, or only text, and so on), in order to identify the best way to reach tourists and citizens in all countries.

## 5. Discussion

The identification of similarities, differences, and peculiarities in a specific area entails different paths toward the development of a smart cross-border tourism destination. Too many similarities across the whole region means that there is relatively little potential for mutual learning. On the other hand, too many differences in competences and characteristics means fewer possibilities for building synergies across the region.

There are two main innovative paths to follow based on the extent of similarities and differences:

(1) Knowledge sharing and transfer—consisting of innovation diffusion, by transmitting existing technologies, products, services, and best practices from one side of the border to another one [66,67], especially when there are significant differences among countries.

(2) Knowledge creation and cooperation—consisting of resource and capability recombination to create novel concepts, competencies, products, and services in order to exploit the unexplored potentialities, especially in the case of similarities among countries.

Differences in competencies and culture act as an enriching and facilitating factor for creating a cross-border strategic brand [68].

However, it is worth noting that such innovation paths have to be fine-tuned to the local condition and reshaped when adapted in other contexts [68]. To this aim, the macro-dimension analysis allowed depicting a picture for all the countries, as well as highlighting that tourism is one of the sectors that contributes the most to the total GDP and employment for each country and that it is possible to intertwine a common canvas in order to build common strategies and services thanks to the similarities between countries. The analysis of TTCIs and the use of the Global Innovation Index demonstrated that the three countries are ready from a smart-configuration point of view, and that the diffusion of technologies is quite anchored within them.

Moreover, this framework provides insights for managers, as it allows better understanding what tourists' needs are, how to ameliorate their services or create new ones, and how they might tackle their marketing strategies for business promotion. In this sense, this methodology offers the opportunity for individual countries of the area to promote the offer of a destination under the unique Adriatic–Ionian brand. This is possible if, first of all, the government and local authorities assume a new role for the support of local private actors and trade associations in the social system and in the promotion of the territory, encouraging and facilitating the creation of an integrated tourism system [12]. An effective territorial promotion cannot fail to take into account the smart technologies and the most widespread means of communication, such as social media. In fact, social media and other OTA channels allow communications to a large number of travellers, showing the characteristics of the territory, its products, and services and partially creating in the potential tourist's mind an anticipation of the future holiday experience [69]. From a literature analysis, it is known that ICT has overhauled the tourism industry, which impacts the way tourism organizations conduct business and interact with their stakeholders, and that technological advances yield major changes in tourism by enabling tourism actors to create markets, management practices, and new competitive strategies. In other words, technologies are transforming the static and practical aspects of the management of tourism and marketing into a dynamic process (in which managers and tourists use technology as a tool) that allows market players and actors (tourism providers, stakeholders, intermediaries, and tourists) in the tourism industry to shape technology and also be affected by it [24]. In fact, during and after the COVID-19 pandemic, digital technologies have assumed a fundamental role as in many sectors [70–73], also in the strategies restarting tourism companies as well as for governments involved in the planning and execution of public policies for sustaining the competitiveness of tourism companies and destinations and managing the crisis, enabling a process of transformation able to reduce the negative impacts of the emergence by creating opportunities for future development [74].

For this purpose, passing through the more detailed analysis made by the administration of questionnaires in the micro-dimension analysis has allowed evaluating in more detail the social traits of the countries and the actual use of technologies, so as to finally pick up those innovative and smart elements that could help the final destinations to be smarter. It is known that the causal conditions affecting the development of smart destinations are the following: increasing internet penetration rate, the use of information and communication technologies, the emergence of the smart city, the development of social networks, and global changes. Thus, among the different issues, the questionnaires were aimed at quantifying the level of use of technologies, the internet, and social media by service providers, intermediaries, and so on, so as to measure how much modern technologies in tourism destinations are used to influence experiences and increase the competitiveness of destinations and projects of tourism development, and to understand the state of the art of the Ionic–Mediterranean area. In spite of the high potential of smart tourism to provide better services to tourists, results have shown that the use of this technology has not been adequately addressed in all countries. Nevertheless, a result highlighted from the micro-analysis is related to the awareness of the potential use that data gathered

from online reviews or social media or through their own sites could generate, from the increment of the competition power among rivals, the creation of personalized services or of marketing strategies, all aspiring towards an improvement in their own competitiveness. In addition, the proposed framework aims to give support to the policy makers and practitioners in order to highlight strengths and weaknesses, cross-correlating distinct and diversified tourism products, linking them with the tourist offerings and valorisation of natural and cultural assets as well as landscapes, traditions, folks, and crafts from a broader perspective, and providing hints for marketing and managing the Adriatic–Ionian area as an inter-regional STD. From this perspective, the use of business analytics has allowed gathering data from the web and inspecting the final destination image from the customer point of view, highlighting the way in which each country and region are perceived and how some countries should improve their marketing and communication strategies in order to finally build a smart cross-border tourism destination. Finally, a mandatory step for the improvement in both a local and inter-regional tourism system might be promoting the engagement of partners, local citizens, and tourists for co-creating a tourism experience. This might ensure a diversification of the offer and an easy adaptation to the needs of tourists [27]. The adoption of this methodological framework will be useful within the process of the development of different geographical areas with a similar vocation but dissimilar characteristics, as the variegated kinds of analysis allow pointing out the main features and needs of each country.

## 6. Conclusions

The conceptual methodological framework presented here integrates different levels of analysis at a macro-, micro-, and customer level into different features that characterize an STD, to evaluate its promptness to adopt a smart configuration. In order to well-characterize a tourism destination from the different perspectives, the combination of several parameters is necessary to have a complete picture of the destination. This framework is conceived as a preliminary action supporting the definition of a strategic roadmap for the development of an STD.

In particular, this framework allows highlighting all the features of each country, from the efficiency of the ICT infrastructures to their peculiarities, drawing attention on the economic factors and on the different tourism national/regional policies and strategies, thus pointing out the strengths, weaknesses, similarities, differences, and peculiarities of each nation or particular region. The framework has been developed in the frame of the NEST (networking for smart-tourism development) Project, a cooperation project aimed towards promoting a sustainable and smart development path in the Ionian–Adriatic macro-region, a cross-border area including regions with a recognized tourism vocation. The framework is a preliminary result of the project, and it has been developed as an investigative tool able to identify all the characteristics of the countries of the area, with the goal of proposing a roadmap for developing a smart cross-tourism destination. Another reason why adopting this framework is vital in this context lies in the fact it allows investigating if and how the tourism destination has an ecosystem conception, by studying how much all the stakeholders are integrated within the development of the STD and the absorption of smartness by all the actors—the country's competitiveness in tourism being a major consideration for policy makers and a major challenge for professionals in providing evidence to inform decision making. Finally, the methodological framework reflects the possibility of implementing new and well-fine-tuned services to the local condition for citizens and tourists, and attempts to provide hints among the tourist actors toward the development of a sustainable tourism destination and the improvement in the customer perception of a destination. In this way, processes of knowledge sharing and transfer and knowledge creation and cooperation are easily allowed in order to drive the creation of a cross-border strategic brand.

As future research directions, we suggest the three main issues that need to be further addressed as follows. First, it is important to explore the impact of ICT, mobile communication,

cloud computing, artificial intelligence, and virtual reality on developing smart tourism [75]. Secondly, it is of great interest to investigate the interaction between smart tourism and sustainability. Lastly, a critical issue is also to evaluate the collaboration and integration of four main actors, that of the quadruple helix (government–academia–business–civil society), that can support the development of smart tourism with smart strategies.

**Author Contributions:** The authors confirm their contributions to the paper as follows: conceptualization, V.N., E.H. and Y.M.; methodology, V.N. and E.H.; formal analysis, Y.M.; investigation, Y.M.; data curation, Y.M.; writing—original draft preparation, V.N., E.H. and Y.M.; writing—review and editing, V.N., E.H. and Y.M.; visualization, V.N., E.H. and Y.M.; supervision, V.N. and E.H.; project administration, V.N.; funding acquisition, V.N. All authors have read and agreed to the published version of the manuscript.

**Funding:** This paper provides a contribution to the research related to the ongoing research project NEST—networking for smart-tourism development—Project n. 96-1° call for standard project—co-financed by the European Union under the Instrument for Pre-Accession Assistance (IPA II).

**Institutional Review Board Statement:** Not applicable.

**Informed Consent Statement:** Not applicable.

**Data Availability Statement:** Not applicable.

**Acknowledgments:** We acknowledge the financial support from the on-going research project NEST—networking for smart-tourism development—Project n. 96-1° call for standard project—co-financed by the European Union under the Instrument for Pre-Accession Assistance (IPA II).

**Conflicts of Interest:** The authors declare no conflict of interest.

## Appendix A

### Appendix A.1. Tourism Economic Impact

The study of the economic impact of the tourism sector is important for understanding the contribution of the sector in the economy of a country. This sector creates jobs, drives exports, and generates prosperity in the world, and thus has a fundamental role in the analysis of the macro-dimension. The analysis of the economic impact of tourism could be performed through the following key indicators:

- Tourism's direct contribution to GDP, related to the internal spending on travel and tourism, by residents and nonresidents and by government "individual" spending for services directly linked to visitors;
- Total indirect contribution to GDP, which includes the related impacts (indirect and induced) and refers to investment activity, such as the purchase of new aircraft and the construction of new hotels, and government "collective" spending which helps tourism activity in many ways, such as tourism marketing and promotion, aviation, administration, and resort area sanitation services;
- Tourism's total contribution to GDP, which includes its wider impacts on the economy (i.e., the indirect and induced impacts), in addition to direct impacts;
- Tourism's total contribution to employment, which includes direct and indirect jobs supported by the travel and tourism industry;
- Capital investment, i.e., investment associated with travel and tourism, both private and public;
- Visitor exports, i.e., spending in the domestic economy by foreign visitors.

The principal data source for these typologies of indicators is represented by the report and statistical data performed by the WTTC organization that annually performs this kind of analysis at international and country levels.

### Appendix A.2. Tourism Statistical Data

The statistical data related to the tourism sector aim to describe the dimension of tourism flows, tourism receipts and expenditure, and the number of tourism enterprises

of the destination to be studied. These kinds of data measure the tourism performance, the ability of a destination to deliver quality and competitive tourism services, and the attractiveness of a destination.

The analysis of the tourism system of a destination could be performed through three main kinds of data:

- Tourism flows;
- Tourism receipts and expenditure;
- Tourism enterprises.

The analysis of tourism flows could be divided into three main categories:

- Domestic tourism, which comprises the activities of a resident visitor within the country observed;
- Inbound tourism is related to the activities of a nonresident visitor within the country observed;
- Outbound tourism includes the activities of a resident visitor outside the country observed.

The assessment of the levels of touristic flows, receipts, and expenditure of a destination takes into account both the arrivals and the nights of visitors and is composed of the following indicators:

---

**TOURISM FLOWS**

**Domestic tourism**
Total domestic trips

<div style="text-align:right">

Overnight visitors (tourists)
Same-day visitors (excursionists)
</div>

Nights in all types of accommodation

<div style="text-align:right">

Hotel and similar establishments
Specialized establishments
Other collective establishments
Private accommodation
</div>

**Inbound tourism**
Total international arrivals

<div style="text-align:right">

Overnight visitors (tourists)
Same-day visitors (excursionists)
Top markets (list)
</div>

Nights in all types of accommodation

<div style="text-align:right">

Hotel and similar establishments
Specialized establishments
Other collective establishments
Private accommodation
</div>

**Outbound tourism**
Total international departures

<div style="text-align:right">

Overnight visitors (tourists)
Same-day visitors (excursionists)
</div>

Top destinations (list)

**TOURISM RECEIPTS AND EXPENDITURE**

**Inbound tourism**
Total international receipts
**Outbound tourism**
Total international expenditure

---

The analysis of tourism enterprises completes the study of the tourism system in a destination and refers to assessing the number and the typologies of companies in the tourism industry.

| TOURISM ENTERPRISES | |
| --- | --- |
| **Tourism Industries** | |
| Accommodation services for visitors | |
| Hotel and similar establishments (with typologies) and number of stars | |
| Food- and beverage-serving industry | |
| Passenger transport | |
| | Air passenger transport |
| | Railway passenger transport |
| | Road passenger transport |
| | Water passenger transport |
| Transport equipment rental | |
| Travel agencies | |
| Cultural industries | |
| Sport and recreation industries | |
| Retail trade of a country | |
| Other related tourism industry | |

These kinds of statistical data are available from the desk analysis of different reports performed by the OECD and also from the official tourism observatory of the destination.

*Appendix A.3. Tourism Competitiveness Indicators*

The T&T Competitiveness Index (TTCI), defined by the WTTC, has adopted the most complete and modern set of indicators globally available to measure tourism competitiveness and it is the most methodological framework useful to describe the macro-dimension of a tourism destination. For its completeness and integration of various elements useful for measuring the level of competitiveness and smartness of a destination, it has been selected as a method of analysis of the physical and infrastructure layer. TTCIs consist of four sub-indices:

- Enabling environment;
- T&T policies and enabling conditions;
- Infrastructure;
- Natural and cultural resources.

These 4 sub-indices are made up of 14 pillars, calculated on the basis of data derived from the executive opinion survey (survey) and quantitative data from other sources; then, each pillar is composed using different variables.

Starting with the TTCI framework, the macro-dimension analysis of a destination considers some variables of those pillars that impact the smartness of the destination: ICT readiness, prioritization of travel and tourism, air transport infrastructure, ground and port infrastructure, tourist service infrastructure, natural resources and cultural resources, and business travel. In the following table, the pillars, sub-indices, and variables useful for macro-dimension analysis are summarized.

| TTCI Index | |
| --- | --- |
| **Index at pillar level** | **Variables** |
| Business environment | |
| Safety and security | |
| Health and hygiene | |
| Human resources and labour market | |
| ICT readiness | ICT use for biz-to-biz transactions<br>Internet use for biz-to-consumer transactions<br>Internet users % pop.<br>Fixed-broadband internet subscriptions/100 pop.<br>Mobile-cellular telephone subscriptions/100 pop.<br>Mobile-broadband subscriptions/100 pop.<br>Mobile network coverage % pop.<br>Quality of electricity supply |
| Prioritization of travel and tourism | Government prioritization of travel and tourism industry (T&T); government expenditure % government budget; effectiveness of marketing and branding to attract tourists; comprehensiveness of annual T&T data 0–120 (best); timeliness of providing monthly/quarterly T&T data; country brand strategy rating |
| International openness | |
| Price competitiveness | |
| Environmental sustainability | |
| Air transport infrastructure | Quality of air transport infrastructure<br>Available seat kilometres, domestic (millions)<br>Available seat kilometres, international (millions)<br>Aircraft departures/1000 pop.<br>Airport density, airports/million pop.<br>Number of operating airlines |
| Ground and port infrastructure | Quality of roads<br>Road density % total territorial area<br>Paved-road density % total territorial area<br>Quality of railroad infrastructure<br>Railroad density, km of roads/land area<br>Quality of port infrastructure<br>Ground transport efficiency |
| Tourist service infrastructure | Hotel room number/100 pop.<br>Quality of tourism infrastructure<br>Presence of major car rental companies<br>Automated teller machines |
| Natural resources | Number of World Heritage natural sites (number of sites)<br>Total known species (number of species)<br>Total protected areas (% total territorial area)<br>Natural tourism digital demand, 0–100 (best)<br>Attractiveness of natural assets |
| Cultural resources and business travel | Number of World Heritage cultural sites (number of sites)<br>Oral and intangible cultural heritage (number of expressions)<br>Sports stadiums (number of large stadiums)<br>Number of international association meetings, 3-year average<br>Cultural and entertainment tourism digital demand |

*Appendix A.4. National and Local Tourism Strategy*

The analysis of the macro-dimension of a tourism destination could be completed through the study of political strategies and programs related to the development of tourism in a smart and sustainable way. The central argument is to see how and to what extent the institutional capacity for coordination, collaboration, and cooperation can be efficiently used as a governance practice (the efficiency of governance) to improve tourism destination competitiveness, helping to transform tourists' needs into solutions and opportunities for smart, inclusive, and sustainable growth.

The study can start with the observation of the different public authorities (minister of tourism, destination's management organization at the national and regional level, and public tourism association) that have the mission to manage the tourism destination.

| GOVERNANCE | |
|---|---|
| Name of the institutions responsible for the management and promotion of the tourism sector, at the national level | e.g., Ministry of Tourism, Ministry of Tourism and Environment, etc. |
| Name of the institutions responsible for the management and the promotion of the tourism sector, at the regional level | e.g., PugliaPromozione, etc. |
| Presence of a tourism development plan/strategy, at the national level | e.g., "Italia Paese per Viaggiatori", etc. |
| Principles of the tourism strategic plan, at the national level | |
| List of principles | e.g., "Sustainability", "Accessibility", " Innovation", etc. |
| Strategic objectives at the national level | |
| List of target objectives | e.g., "Bost the tourism system's competitiveness". |
| Presence of development/strategic plan for tourism, at the local level | e.g., "Puglia 365", etc. |
| Thematic area of development/strategic plan for tourism, at the local level | |
| List of thematic areas | e.g., "Hospitality", "Education", "Infrastructure", etc. |
| Presence of destination management organization systems at national level | Name of the DMS at the national Level, e.g., "Enit.it2 |
| Presence of destination management organization systems at regional level | Name of the DMS at the local level, e.g., "agenziapugliapromozione.it" |
| Public organization or public private partnership to promote the tourism destination | List of other public or public/private organizations for tourism management |

*Appendix A.5. Destination Identity/Brand*

The final kind of analysis that could be performed for the study of a destination from a macro-dimension perspective is related to business analytics on social media related to the destination brand (official hashtags or keywords). This kind of analysis requires different phases:

- Identification of an official hashtag or keyword to monitor, related to a specific destination;
- Identification of social media analytical tools;
- Identification of an official social network account (Facebook, Instagram, or Twitter);
- Definition of the observation time;
- Identification of social media metrics to analyse;
- Identification of social media metrics related to a social media account (Facebook, Instagram, or Twitter).

**Appendix B.**

*Questionnaire for Intermediaries*

1.

| I Am a: | Name of Intermediary | Ownership | Website (If Yes, Indicate the Name) | Mobile App (If Yes, Indicate the Name) |
|---|---|---|---|---|
| ☐ Tour operator | | ☐ Public | __________ | __________ |
| ☐ Travel agent | | ☐ Private | __________ | __________ |
| ☐ Other _______ | | ☐ Other _______ | __________ | __________ |

2. **What are the reasons of usage of internet? Indicate from 1 (low) to 7 (high).**

| | Not Used at All | 1 | 2 | 3 | 4 | 5 | 6 | 7 |
|---|---|---|---|---|---|---|---|---|
| 1. Managing requests of info from customers | ☐ | ☐ | ☐ | ☐ | ☐ | ☐ | ☐ | ☐ |
| 2. Managing promotions and special offerings | ☐ | ☐ | ☐ | ☐ | ☐ | ☐ | ☐ | ☐ |
| 3. Managing customers and fidelity initiatives | ☐ | ☐ | ☐ | ☐ | ☐ | ☐ | ☐ | ☐ |
| 4. Market researches | ☐ | ☐ | ☐ | ☐ | ☐ | ☐ | ☐ | ☐ |
| 5. Buying of products and services | ☐ | ☐ | ☐ | ☐ | ☐ | ☐ | ☐ | ☐ |
| 6. Communicate with public administrations | ☐ | ☐ | ☐ | ☐ | ☐ | ☐ | ☐ | ☐ |
| 7. Managing relationships with touristic firms and association categories | ☐ | ☐ | ☐ | ☐ | ☐ | ☐ | ☐ | ☐ |
| 8. Monitoring and management of online feedbacks | ☐ | ☐ | ☐ | ☐ | ☐ | ☐ | ☐ | ☐ |
| 9. Online payment | ☐ | ☐ | ☐ | ☐ | ☐ | ☐ | ☐ | ☐ |

3. **What kind of technologies are used for marketing purposes? Indicate from 1 (low) to 7 (high).**

| | Not Used at All | 1 | 2 | 3 | 4 | 5 | 6 | 7 |
|---|---|---|---|---|---|---|---|---|
| 1. Telephone | ☐ | ☐ | ☐ | ☐ | ☐ | ☐ | ☐ | ☐ |
| 2. E-mail | ☐ | ☐ | ☐ | ☐ | ☐ | ☐ | ☐ | ☐ |
| 3. Web site | ☐ | ☐ | ☐ | ☐ | ☐ | ☐ | ☐ | ☐ |
| 4. Facebook | ☐ | ☐ | ☐ | ☐ | ☐ | ☐ | ☐ | ☐ |
| 5. Twitter | ☐ | ☐ | ☐ | ☐ | ☐ | ☐ | ☐ | ☐ |
| 6. App | ☐ | ☐ | ☐ | ☐ | ☐ | ☐ | ☐ | ☐ |
| 7. Other online tools | ☐ | ☐ | ☐ | ☐ | ☐ | ☐ | ☐ | ☐ |

4. **What is the level of use of the following technologies? Indicate from 1 (low) to 7 (high).**

| | Not Used at All | 1 | 2 | 3 | 4 | 5 | 6 | 7 |
|---|---|---|---|---|---|---|---|---|
| 1. Systems for users profiling | ☐ | ☐ | ☐ | ☐ | ☐ | ☐ | ☐ | ☐ |
| 2. Virtual guides, interactive maps, 3D, rebuilding, augmented realities | ☐ | ☐ | ☐ | ☐ | ☐ | ☐ | ☐ | ☐ |
| 3. Digital marketplaces (Booking, Expedia, and so on) | ☐ | ☐ | ☐ | ☐ | ☐ | ☐ | ☐ | ☐ |
| 4. Mobile Application | ☐ | ☐ | ☐ | ☐ | ☐ | ☐ | ☐ | ☐ |
| 5. CRM systems | ☐ | ☐ | ☐ | ☐ | ☐ | ☐ | ☐ | ☐ |
| 6. Social network, blog, forums | ☐ | ☐ | ☐ | ☐ | ☐ | ☐ | ☐ | ☐ |

5.  **What is the level of use of the following technologies? Indicate from 1 (low) to 7 (high).**

| | Not Used at All | 1 | 2 | 3 | 4 | 5 | 6 | 7 |
|---|---|---|---|---|---|---|---|---|
| 1. Systems for users profiling | ☐ | ☐ | ☐ | ☐ | ☐ | ☐ | ☐ | ☐ |
| 2. Virtual guides, interactive maps, 3D, rebuilding, augmented realities | ☐ | ☐ | ☐ | ☐ | ☐ | ☐ | ☐ | ☐ |
| 3. Digital marketplaces (Booking, Expedia, and so on) | ☐ | ☐ | ☐ | ☐ | ☐ | ☐ | ☐ | ☐ |
| 4. Mobile Application | ☐ | ☐ | ☐ | ☐ | ☐ | ☐ | ☐ | ☐ |
| 5. CRM systems | ☐ | ☐ | ☐ | ☐ | ☐ | ☐ | ☐ | ☐ |
| 6. Social network, blog, forums | ☐ | ☐ | ☐ | ☐ | ☐ | ☐ | ☐ | ☐ |

6.  **Where do you save the gathered data?**

    [ ] Data warehouse
    [ ] Cloud
    [ ] Distributed data stores
    [ ] Other, specify________________________________________________________

7.  **How do you think all your data could be used for?**

    [ ] Increase competition power among rivals
    [ ] Create personalized services for customers
    [ ] Discover patterns and trends
    [ ] Prepare demand forecasting
    [ ] Prepare dynamic context and pricing
    [ ] Prepare marketing strategies
    [ ] Improve competitiveness
    [ ] Improve smart hospitality
    [ ] Other, specify________________________________________________________

8.  **What kind of tools are used by your company to monitor customer satisfaction? Indicate from 1 (low) to 7 (high).**

| | Not Used at All | 1 | 2 | 3 | 4 | 5 | 6 | 7 |
|---|---|---|---|---|---|---|---|---|
| 1. Questionnaires | ☐ | ☐ | ☐ | ☐ | ☐ | ☐ | ☐ | ☐ |
| 2. Email | ☐ | ☐ | ☐ | ☐ | ☐ | ☐ | ☐ | ☐ |
| 3. Feedback/reviews | ☐ | ☐ | ☐ | ☐ | ☐ | ☐ | ☐ | ☐ |
| 4. Forum | ☐ | ☐ | ☐ | ☐ | ☐ | ☐ | ☐ | ☐ |
| 5. Blog | ☐ | ☐ | ☐ | ☐ | ☐ | ☐ | ☐ | ☐ |
| 6. CRM system | ☐ | ☐ | ☐ | ☐ | ☐ | ☐ | ☐ | ☐ |
| 7. Software for statistical analysis | ☐ | ☐ | ☐ | ☐ | ☐ | ☐ | ☐ | ☐ |
| 8. Social networks | ☐ | ☐ | ☐ | ☐ | ☐ | ☐ | ☐ | ☐ |
| 9. Traditional tools (guest book, and so on) | ☐ | ☐ | ☐ | ☐ | ☐ | ☐ | ☐ | ☐ |
| 10. Vis-à-vis | ☐ | ☐ | ☐ | ☐ | ☐ | ☐ | ☐ | ☐ |

9.  **For which type of the following activities do you collaborate with other partners?**

    [ ] Promotion/exploitation of the regional tourist offer
    [ ] Training activities
    [ ] Participation at public or private call for funding
    [ ] Joint services of consulting
    [ ] Creation of an integrated tourist system
    [ ] Co-working activities

[ ] Co-marketing and co-branding activities
[ ] Creation of new tourist product/services
[ ] Creation of regional brand
[ ] Activities of sensitization to adopt ICTs
[ ] Other specify _______________________________________________

**Are they a benefit for your company? □ Y □ N**

10. **Do you have any kind of collaboration/interaction with customers? □ Y □ N. If yes, what kind?**

[ ] In the activities of Marketing & Sales
[ ] In the activities of Customer Care
[ ] In the activities of new product development
[ ] Other, specify_______________________________________________

11. **What are the advantages of social network adoption?**

[ ] Increase total customers
[ ] Increase visibility
[ ] Provide multi-channel support
[ ] Cost reduction
[ ] Other, specify_______________________________________________

12. **Do you have a collaboration with other partners, in order to incentive the touristic tours in the city? □ Y □ N. If yes, what kind of partner do you have?**

[ ] Hotels, B&Bs, agritourists
[ ] Private companies compatible with sustainable tourism
[ ] Culinary, farm, restaurant
[ ] Artist and artisans
[ ] Local associations
[ ] Other, specify_______________________________________________

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
