# Peer review of "A Methodological Framework for Developing a Smart-Tourism Destination in the Southeastern Adriatic–Ionian Area"

_sustainability, doi:10.3390/su15032057_

Round 1
Reviewer 1 Report
The study is very well carried out and structured. This work is very useful in the field of smart tourism. I am not the best person to assess the English writing but I was able to read the article in a very comprehensive and understandable way what makes me think that the written is good.
However, it should point out some of the conclusions, implications and research gaps. Perhaps it deserved more bibliographic references related to international studies. See the following articles on intentions to intervene in the environment that should be added to your work, taking into account the need for more interventions to respond to climate change problems namely with smart strategies, particularly in urban areas (https://doi.org/10.1016/j.jenvman.2022.115161), during the COVID-19 period (https://www.mdpi .com/2071-1050/13/11/6399) and identified challenges for the tourism sector globally (https://doi.org/10.1016/j.seta.2021.101494).
Congratulations. Good Work!
Author Response
Dear Reviewer,
Thank you for your positive approach toward our work, we appreciate it. Thank you also for your valuable comments since we had the opportunity to read the suggested works and add to our study as well. We added the following references among others.
ÖdemiÅŸ, M. (2022). Smart Tourism Destinations: A Literature Review on Applications in Turkey's Touristic Destinations. Optimizing Digital Solutions for Hyper-Personalization in Tourism and Hospitality, 131-153.
EÅŸitti, B. B. (2021). COVID-19 and Alternative Tourism: New Destinations and New Tourism Products. In Handbook of Research on the Impacts and Implications of COVID-19 on the Tourism Industry (pp. 786-805). IGI Global.
Lopes, H. S., Remoaldo, P. C., Ribeiro, V., & Martín-Vide, J. (2022). Pathways for adapting tourism to climate change in an urban destination–Evidences based on thermal conditions for the Porto Metropolitan Area (Portugal). Journal of Environmental Management, 315, 115161.
Lopes, H. S., Remoaldo, P. C., Ribeiro, V., & Martín-Vide, J. (2021). Effects of the COVID-19 pandemic on tourist risk perceptions—The case study of Porto. Sustainability, 13(11), 6399.
Lu, C. W., Huang, J. C., Chen, C., Shu, M. H., Hsu, C. W., & Bapu, B. T. (2021). An energy-efficient smart city for sustainable green tourism industry. Sustainable Energy Technologies and Assessments, 47, 101494.
Rodrigues, J. M., Cardoso, P. J., Monteiro, J., & Ramos, C. M. (2020). Augmented intelligence: Leverage smart systems. In Smart Systems Design, Applications, and Challenges (pp. 1-22). IGI Global.
Reviewer 2 Report
Review article: A methodological framework for developing a smart tourism destination in the South-Eastern Adriatic-Ionian area.
By: Valentina Ndou, Eglantina Hysa
Line: 91-92 (Xiang & Fesenmaier, 2017; Bănescu et al., 2021: Gretzel et al., 2015; Ndou et al., 2022). Different letters.
Line 107: dation for the birth of the Smart tourism destination (STD) where technology becomes an – the whole shortcut should be written with big letters
In point: 2. Background it is a lot about STD but it is not enough information about tourism and especially about smart tourism also with definition and methodology.
Line 163: cultural/historic heritage – it would be enough just cultural heritage – because historic it is a part of cultural heritage.
Figure 4 it’s not clear – the date bases are not able to read – the same with Figure 5.
The article should start from idea from SMART destination – and describes different part of tourism.

Author Response
Dear Reviewer,
Thank you for your positive approach toward our work, we appreciate it. Thank you also for your valuable comments since we had the opportunity to rearrange and correct some issues. As such, please find below the comments for the issues raised. You may also find the updated version of the paper with track changes. Thank you again!
- Review article: A methodological framework for developing a smart tourism destination in the South-Eastern Adriatic-Ionian area.
By: Valentina Ndou, Eglantina Hysa, Ylenia Maruccio
Line: 91-92 (Xiang & Fesenmaier, 2017; Bănescu et al., 2021: Gretzel et al., 2015; Ndou et al., 2022). Different letters.
This was revised and rechecked with the reference list.
- Line 107: dation for the birth of the Smart tourism destination (STD) where technology becomes an – the whole shortcut should be written with big letters
Thank you, it has been corrected.
- In point: 2. Background it is a lot about STD but it is not enough information about tourism and especially about smart tourism also with definition and methodology.
We added some information on page 3. Thank you!
- Line 163: cultural/historic heritage – it would be enough just cultural heritage – because historic it is a part of cultural heritage.
Rearranged, thank you!
- Figure 4 it’s not clear – the date bases are not able to read – the same with Figure 5.
Thank you for your suggestion, all Figures, from 4th to 9th, have been ameliorated.
- The article should start from idea from SMART destination – and describes different part of tourism.
Thank you for the comment. We followed your suggestions and explained the evolution of concepts better in the paper.
Reviewer 3 Report
The authors do not explain how research is a step forward for scientific knowledge.
I recommend the authors to consider in the future articles the papers considered relevant, especially those in the main flow of publications.
Author Response
Dear Reviewer,
Thank you for your positive approach toward our work, we appreciate it. Thank you also for your valuable comments since we had the opportunity to rearrange and correct some issues. As such, please find below the comments for the issues raised. You may also find the updated version of the paper with track changes. Thank you again!
- The authors do not explain how research is a step forward for scientific knowledge.
Some additional information is added in the introduction session. Furthermore, some more references on the support of our work.
In specific, you can find it at lines 72-78.
- I recommend the authors to consider in the future articles the papers considered relevant, especially those in the main flow of publications.
Some additional information is added in the conclusion session.
Thank you for your advices!
Round 2
Reviewer 2 Report
Non